# Aberrant Notch Signaling Pathway as a Potential Mechanism of Central Precocious Puberty

**DOI:** 10.3390/ijms23063332

**Published:** 2022-03-19

**Authors:** Young Suk Shim, Hae Sang Lee, Jin Soon Hwang

**Affiliations:** Department of Pediatrics, Ajou University School of Medicine, Suwon 164999, Korea; royjays@gmail.com (Y.S.S.); pedhwnag@ajou.ac.kr (J.S.H.)

**Keywords:** central precocious puberty, etiology, puberty, Notch signaling pathway

## Abstract

The Notch signaling pathway is highly conserved during evolution. It has been well documented that Notch signaling regulates cell proliferation, migration, and death in the nervous, cardiac, and endocrine systems. The Notch pathway is relatively simple, but its activity is regulated by numerous complex mechanisms. Ligands bind to Notch receptors, inducing their activation and cleavage. Various post-translational processes regulate Notch signaling by affecting the synthesis, secretion, activation, and degradation of Notch pathway-related proteins. Through such post-translational regulatory processes, Notch signaling has versatile effects in many tissues, including the hypothalamus. Recently, several studies have reported that mutations in genes related to the Notch signaling pathway were found in patients with central precocious puberty (CPP). CPP is characterized by the early activation of the hypothalamus–pituitary–gonadal (HPG) axis. Although genetic factors play an important role in CPP development, few associated genetic variants have been identified. Aberrant Notch signaling may be associated with abnormal pubertal development. In this review, we discuss the current knowledge about the role of the Notch signaling pathway in puberty and consider the potential mechanisms underlying CPP.

## 1. Introduction

Puberty is one of the key stages in the developmental process between childhood and adulthood. During this period, secondary sexual characteristics emerge, the body rapidly grows, the reproductive system matures, and many social and emotional changes occur [1]. The factors that trigger puberty have not yet been entirely elucidated. Pubertal onset is a complex process that is influenced by genetic and/or environmental factors, such as nutritional status and exposure to endocrine-disrupting chemicals [2,3,4,5]. Genetic factors are thought to play an essential role in puberty initiation. The onset of puberty is different among races/ethnic groups, and the similarity of menarcheal age between mothers and daughters further supports that genetic variations influence the beginning of puberty. Moreover, the concordance of pubertal timing is greater in monozygotic than in dizygotic twins [6]. Genetic factors are thought to account for 50–80% of the variation in the onset of puberty [7,8]. However, despite the strong heritability of pubertal onset and progression, our knowledge of the underlying genetic background remains limited.

To date, many researchers have tried to identify the pathogenic genes that cause central precocious puberty (CPP). However, monogenic causative mutations have been identified in only four genes, namely, *KISS1*, Kiss1 receptor (*KISS1R*), Makorin Ring Finger Protein 3 (*MKRN3*) and Delta-Like Homolog 1 (*DLK1*) [9]. These findings suggest the presence of genetic heterogeneity in the onset of puberty and emphasize the need for further genetic studies. Most recently, a paternally inherited deletion of *DLK1*, a noncanonical ligand that binds to Notch receptors, has been identified in several family members with CPP [10]. The Notch signaling pathway is highly conserved from Drosophila to humans. It has been well described that Notch signaling pathways play fundamental roles in the regulation of cell proliferation, migration, and death in the nervous, cardiac, and endocrine systems [11]. Moreover, Notch signaling plays a critical role in the cell fate of hypothalamic Kisspeptin neurons [12].

Therefore, we summarize the current knowledge of the normal physiology of pubertal onset and discuss whether the Notch signaling pathway is a potential regulator of pubertal onset and progression.

## 2. Physiology of Puberty: Kisspeptin/Neurokinin B (NKB)/Dynorphin (Dyn) Neuropeptides

Secondary sexual development, including breast engorgement or testicular enlargement, is initiated through reactivation of the hypothalamus–pituitary–gonadal (HPG) axis, after the mini puberty during the fetal and early postnatal periods [13]. Prior to the onset of puberty, the levels of gonadotropins such as luteinizing hormone (LH) and follicle stimulating hormone (FSH) are not elevated, despite the low serum sex hormone levels due to the negative feedback mechanism. The HPG axis acts in an orchestrated manner to modulate the neurosecretory activity of gonadotropin-releasing hormone (GnRH) neurons [14]. When the HPG axis is reactivated, the pulsatile GnRH release re-emerges, possibly due to an increase in activators such as kisspeptin signals, which stimulate GnRH neurons to control the pulsatile GnRH release. This occurs via the elevation of LH and FSH levels through the activity of the pituitary, with the subsequent downstream release of sex hormones, including estrogen and testosterone [15]. In the early stage of puberty, LH secretion begins to increase at night, and its levels gradually increase in amplitude and frequency, promoting gonadal development and sex hormone secretion. During mid-puberty, pulsatile LH secretion occurs during the day, and the secretion cycle is 90–120 min [16].

*KISS1* neurons producing kisspeptin are key players in neuronal networks to regulate pulsatile GnRH secretion (Figure 1). Kisspeptin, a peptide hormone expressed in the hypothalamus and encoded by *KISS1*, binds to the *KISS1* receptor encoded by *KISS1*R [17]. *KISS1* was first discovered as a tumor metastasis suppressor in 1996 and is located in the long arm of chromosome 1q32. The gene contains three exons, of which two are partially translated (exons 2 and 3), including a 145-amino acid precursor peptide [18]. There are two major populations of hypothalamic kisspeptin neurons. One population is primarily located in the infundibular region in the arcuate nucleus (ARC), whereas the other is found in more rostral regions, in the anteroventral periventricular nucleus (AVPV) [15]. The ARC *KISS1* neuronal population is involved in mediating the negative feedback to the actions of sex steroids, such as estrogen, and plays an essential role in the control of the pulsatile secretion of gonadotropins. In addition, AVPV *KISS1* neurons are responsible for the positive feedback control of estrogen release, which drives the preovulatory surge of gonadotropins [19].

A large subset of the ARC neurons, excluding the AVPV *KISS1* neuronal population, produce the neuromodulators NKB and Dyn [20]. Loss-of-function mutations in the tachykinin 3 (TAC3) gene, which encodes the stimulatory neuromodulator NKB, and tachykinin receptor 3 (TACR3), which encodes the NKB receptor, are associated with the development of hypogonadism and delayed puberty. NKB stimulates arcuate *KISS1* neurons, which project to GnRH nerve terminals in the median eminence, and thereby enhance GnRH release [21]. Kisspeptin/NKB-producing neurons in ARC co-express Dyn, an endogenous opioid peptide that inhibits kisspeptin secretion. In animal studies, the administration of Dyn receptor antagonists led to early pubertal onset [22]. The intricate relationship between kisspeptin, NKB, and Dyn, often collectively referred to as KNDy, co-ordinate the pulsatile GnRH secretion in an autocrine/paracrine manner. NKB initiates and/or stimulates synchronized neuronal activity through the stimulatory NKB receptors to release kisspeptin, whereas Dyn exerts an inhibitory action through inhibitory opioid receptors [23].

Thus, ARC and AVPV *KISS1* neurons contribute differently to the control of the pulse and surge patterns of gonadotropin secretion, and are differentially regulated by estrogen. In addition, these neurons are endowed with a diverse set of neuromodulators that synergize with kisspeptin in the control of pubertal onset and progression.

## 3. Central Precocious Puberty

CPP is defined as early activation of the hypothalamic–pituitary–gonadal (HPG) axis. Girls and boys with CPP show progressive sexual development, such as breast development and testicular enlargement before 8 and 9 years of age, respectively, and have elevated gonadotropin levels [24]. In a Korean epidemiological study, the incidence of CPP was found to be 122.8 per 100,000 children, with the incidence being higher in girls (262.8) than in boys (7.0). The estimated prevalence of CPP in Western countries is approximately 1 in 5000 to 10,000 people [14,25,26]. Idiopathic CPP, defined as CPP with no specific organic cause, is found in 90% to 95% of girls and 50% of boys with CPP, respectively. Although the mechanism underlying idiopathic CPP is not clear, it is thought that it involves a complex interplay of genetic and environmental factors. In particular, it has been reported that genetic factors influence CPP development in 50–80% of CPP cases [27].

When puberty begins, pulsatile LH secretion is observed during sleep, and when GnRH is intravenously administered, the LH secretion response increases. In CPP, a rapid growth rate, secondary sexual appearance before 8 years in girls and 9 years in boys, and advanced bone age are observed. To confirm the activation of the HPG axis, it is necessary to perform a GnRH stimulation test. For the GnRH stimulation test, GnRH releasing hormone (100 mcg) is administered intravenously, and LH and FSH levels are continuously measured at 15–30 min intervals for up to 90–120 min. When peak LH levels during GnRH stimulation are above 5 IU/L, CPP is diagnosed [24]. As additional diagnostic criteria, a peak LH/FSH ratio of >0.66 has been suggested with a reported sensitivity of 82% and specificity of 97% [28].

The objective of CPP treatment is to achieve normal pubertal development, such as menarche, similar to that in peers, to improve final adult height, and to reduce psychosocial problems [29]. If untreated, it will lead to a final height loss of around 12 and 20 cm in girls and boys, respectively. CPP is treated with a GnRH agonist, which has a longer and stronger action time compared with that of endogenous GnRH. It downregulates GnRH receptor expression and suppresses gonadotropin secretion [30]. Early diagnosis and treatment of CPP is good for prognosis, leading to, for example, an increase in final adult height [31]. However, it is time-consuming and expensive to diagnose CPP because various tests must be conducted. If the causative genes for CPP are further identified in patients with a family history of CPP, personalized treatment can be performed in the future using genetic testing.

## 4. Known Genetic Causes of CPP

### 4.1. KISS1 and KISS1R

In 2008, Teles et al. [32] reported a heterozygous gain-of-function mutation (p. R386P) in *KISS1*R in a Brazilian girl with CPP. She showed Tanner stage IV at the age of 8, a rapid growth rate, and more advanced bone age than those of her chronological age. This gain-of-function mutation increases GnRH secretion and accelerates puberty by prolonging the activation of intracellular *KISS1*R downstream signaling in response to kisspeptin stimulation (Table 1).

Two years after *KISS1*R mutation was identified in a girl with CPP, Silveira et al. [33] reported gain-of-function mutations in *KISS1* in three patients (one boy and two girls) with CPP [28]. The affected boy with the p.P74S mutation in *KISS1* showed early pubertal development at 17 months of age. In functional analysis, the heterozygous mutant (p.P74S) showed degradation resistance and increased Kisspeptin bioavailability. Furthermore, homozygous mutations in *KISS1* (p.H90N) in two unrelated Brazilian girls with CPP were identified. This mutant was not found in healthy controls with normal puberty.

Given the essential functions of *KISS1* and *KISS1*R at pubertal onset, it is conceivable that mutations and polymorphisms in *KISS1* and *KISS1*R may be associated with CPP development. However, so far, mutations in *KISS1* and *KISS1*R have been rarely found in patients with CPP [34,35]. Ko et el. [36] reported eight polymorphisms in *KISS1* in Korean girls with CPP, out of which p.P110T was less frequently detected in patients with CPP than in the controls. Moreover, patients with the p.P110T polymorphism had lower FSH levels during GnRH stimulation than those without it. However, pathological mutations in *KISS1* and *KISS1*R were not found in patients with CPP [37,38].

### 4.2. Makorin Ring Finger Protein 3 (MKRN3)

A mutation in *MKRN3* was first described in several family members with CPP in 2013 [39]. In this study, the authors performed whole exome sequencing in 40 members of 15 families with CPP from several ethnic groups (12 Brazilian, 2 American, and 1 Belgian families). Several variants, such as frameshift, nonsense, and missense mutations (p.A213Gfs*73, p.W391fs*, p.A162Gfs*14, p.R365S), in *MKRN3* were identified. Since the initial report in 2013, *MKRN3* mutations have been reported in various ethnicities and countries [40,41]. *MKRN3* mutations are now the most commonly known genetic factor in CPP. A recent review reported that 48 different *MKRN3* mutations were found in 115 patients with CPP [40]. The prevalence of *MKRN3* mutations was 33–46% in familial and 0.4–3.8% in sporadic CPP [41]. In addition, the frequency of *MKRN3* mutations appears to be higher in boys than in girls with CPP [42]. Furthermore, the frequency of mutations in *MKRN3* has been reported to differ according to ethnic group. Lee et al. [43] detected one novel nonsense mutation (p.Q281*) in *MKRN3* in 260 Korean girls with CPP. This mutation was inherited by the father. The proband revealed accelerated growth, advanced bone age, and pubertal level of LH on the GnRH stimulation test. *MKRN3* mutations have been identified in only 1 in 260 girls with CPP in Korea, a significantly lower frequency than that in Western countries.

The exact mechanisms by which *MKRN3* mutations affect pubertal onset and progression remain to be elucidated. The serum MKRN3 concentrations decreased prior to the onset of puberty, and the circulating MKRN3 levels were negatively correlated with the gonadotropin levels in girls with normal puberty [44]. These findings suggest that *MKRN3* may have an inhibitory effect on the onset of puberty. *MKRN3* is located on chromosome 15q11.2 and consists of three zinc finger domains (C3H), one zinc RING finger domain (C3HC4), and one MKRN3-specific Cys-His domain (CH) [45]. Based on the structure, *MKRN3* is predicted to function as an estimated E3-ubiquitin-linked enzyme, potentially affecting gene expression, targeted proteolysis, and protein function regulation through its E3-ubiquitin-linked enzyme activity [40].

### 4.3. Delta-like Homolog 1 (DLK1)

Dauber et al. [10] detected a paternally inherited large deletion in *DLK1* using whole genome sequencing in a Brazilian family that included five females with CPP. In four affected sisters, the onset of puberty was between the ages of 4.6 and 5.9 years. In a study by Gomes et al. [46], three frameshift mutations in *DLK1* with paternal expression were identified in five female patients with a history of CPP or early menarche. Interestingly, patients harboring *DLK1* mutations showed more metabolic abnormalities, including glucose intolerance, type 2 diabetes mellitus, and obesity, than CPP patients not harboring *DLK1* mutations [46]. Moreover, a rare heterozygous deletion in the splice site junction of *DLK1* was identified in a Spanish girl with no family history of CPP. Her pubertal signs first began at the age of 5.7 years [47]. In Korean girls with CPP, five polymorphisms of *DLK1* were identified, but no high-impact mutations, such as frameshift or nonsense variants, were detected [48]. In another study by Chen et al. [49], no pathogenic *DLK1* mutations were detected in 19 Chinese girls with CPP and early puberty.

*DLK1* is a noncanonical ligand in the Delta-Notch signaling pathway, known to be expressed in the normal pituitary gland and hypothalamus. Furthermore, *DLK1* plays a role in cell differentiation, mostly inhibiting adipocyte differentiation [50,51,52]. However, the neuroendocrine mechanisms underlying the function of *DLK1* in CPP remain unclear. Further studies are needed to elucidate the reproductive function of *DLK1*.

Interestingly, both *MKRN3* and *DLK1* are imprinted genes in which only single-parent alleles are expressed. Epigenetic imprinting disorders have been reported in several human disorders, including Temple syndrome, Prader–Willi syndrome, Beckwith–Bideman syndrome, and Russell–Silver syndrome [53]. Kotler et al. [54] suggested that imprinted genes that promote rapid growth tend to be maternally expressed, while paternally expressed genes tend to delay growth. Furthermore, patients with Temple syndrome are generally thought to have an early onset of puberty. Temple syndrome is the result of uniparental maternal expression of genes on chromosome 14 [55].

## 5. Notch Signaling Pathway

In the past decade, genetic loci or novel candidate genes that affect pubertal timing have been identified with advancements in genetic evaluation techniques, such as genome-wide association studies (GWASs) and next-generation sequencing [56]. Recently, studies have reported that abnormal Notch signaling may be related to CPP development [57].

## 6. Notch Signaling Pathway: Ligands and Receptors

The Notch signaling pathway is a genetically highly conserved signaling cascade that plays an important role in maintaining homeostasis and regulates embryonic development, cell proliferation, and cell death [58]. In mammals, there are four Notch receptors (Notch1–Notch4) and five classic Delta/Serrate/Lag-2 (DSL) ligands: Jagged (Jagged-1, -2) and Delta (Delta-1, -3, -4). These ligands have a DSL region that binds to the Notch receptor [59]. The Notch receptor is a single-pass type I transmembrane protein that is normally present in the cell membrane. Notch receptors contain 29–36 N-terminal epidermal growth factor (EGF) repeats, of which repeats 11 and 12 are essential for ligand binding. When the Notch receptor binds to a ligand, it splits into an extracellular domain (ECD) and intracellular domain (ICD). The Notch ECD includes an EGF-like repeat domain and a Lin12/Notch repeat domain, whereas the Notch ICD includes the RBPJ associated molecule (RAM), ankyrin repeat, TAD (transcription activation), and PEST (rich in proline, glutamate, serine, and threonine) domains (Figure 2). Each of these domains may play an important role in the Notch signaling pathway [60].

Because both polypeptides have a heterodimerization domain in common, they can exist in a heterodimer state. The Notch ICD has a transmembrane domain in contact with the cell membrane and can therefore move freely within the cell. The released intracellular active Notch ICD fragment migrates into the nucleus and binds to CSL (CBF1/recombining binding protein suppressor of hairless (RBPJ-κ)), Suppressor of Hairless (SuH) of Drosophila melanogaster, Lag-1, and mastermind-like (MAML) in Caenorhabditis elegans. When the activated Notch ICD binds to RBPJ-κ through the RAM domain, it affects the transcription of downstream target genes, including those of the HES (Hes/E(spl) family) and HEY (Hesr/Hey) families [58].

The frequency of Notch ligand and receptor binding determines the amount of Notch ICD generated. The amount of Notch ICD that could be potentially generated is affected by the endocytosis of Notch ligands and receptors [61]. Endocytic transfer and the processing of Notch ligands are critical steps for the signaling activity of these molecules. Ligand endocytosis is initiated by ubiquitination mediated by E3 ubiquitin ligases. After endocytosis, a largely unknown process occurs, resulting in a more active cell surface ligands [62]. E3 ligases regulate the Notch signaling cascade by ubiquitination of ligands and Notch receptors, which causes their eventual degradation, or by other ways. Abnormally high or low levels of the Notch ICD may result in disease development.

## 7. Clinical Evidence Demonstrates That Pubertal Onset May Be Associated with the Activity of the Notch Signaling Pathway

By using whole-exome sequencing, we have previously revealed missense variants of *NOTCH2* and *HERC2* in two siblings with CPP [57]. The proband was referred to the hospital at the age of 8.8 years for breast enlargement that started over a year earlier. Physical examination revealed Tanner stage III. On the GnRH stimulation test, the peak LH and FSH levels were 13.4 IU/L and 8.7 IU/L, respectively. The proband’s younger sister was also referred to the hospital because of breast development when she was 8.2 years old. The proband’s sister showed breast development six months earlier than the proband. She showed Tanner stage II, and her bone age was greater than her chronological age. During the GnRH stimulation test, her peak LH level was also above 5 IU/L. The proband and her younger sister had CPP, and they both harbored missense variants in *NOTCH2* and *HERC2*, but their parents, who each had either the NOTCH2 or HERC2 variant, showed no CPP phenotype. *HERC2* is located on chromosome 15q13.1 and encodes a large protein with a predicted molecular mass of over 500 kDa. *HERC2* contains several motifs, including three RCC1-like domains and a C-terminal HECT domain, found in several E3 ubiquitin protein ligases [63]. *HERC2* regulates *LRRK2* and physically interacts with the Notch ligand Dll1/Dl [64]. The protein complex comprising *LRRK2, NEURL4*, and *HERC2* slows the turnover of Dll1 and then negatively modulates the Notch signaling pathway. Furthermore, *HERC2* promotes the differentiation of neural stem cells in mice and regulates the function and survival of mature dopaminergic neurons in adult Drosophila [64]. The authors hypothesized that mutations in *NOTCH2* and *HERC2* might have a synergistic effect on the Notch signaling pathway.

In addition, Giannakopoulos et al. [65] described a boy with Phelan–McDermid syndrome who showed CPP at the age of 1 year. He was a carrier of a 25 kB duplication of the 9q34.3 chromosomal region that included *NOTCH1*. Subjects with Phelan–McDermid syndrome usually appear to develop puberty at normal times [66]. *NOTCH1* duplication may have caused the early onset of puberty in this patient.

## 8. Potential Link between the Notch Signaling Pathway and HPG Axis

The exact mechanism of how the Notch signaling pathway affects the onset of puberty remains unknown. The genes involved in the Notch signaling pathway, including Notch receptors, Dll1, Hes1, and Hey1 are expressed in progenitor cells of the anterior pituitary during postnatal development and in the adult pituitary. The importance of the Notch signaling pathway in the development of kisspeptin neurons is not limited to early neuronal development, but is also required in adulthood. Notch 2 conditional knockout mice showed progressive loss of the progenitor cell population, as well as decreased proliferation during postnatal pituitary development [67]. Moreover, Raetzman et al. [68] reported that persistently activated Notch2 expression results in the delayed appearance of LH and FSH in pituitary gonadotropes. These findings suggest that the Notch signaling pathway is necessary for the maintenance and localization of progenitor cells in the pituitary. Furthermore, aberrant Notch signaling may activate gonadotrope differentiation and function.

The *DLK1* intracellular domain has been shown to negatively regulate Notch signaling by disrupting the RBPJ-κ/Notch signaling pathway [69]. Apart from the canonical ligands, many non-canonical ligands have also been reported to activate or suppress Notch signaling. *DLK1* is a noncanonical ligand that binds to Notch receptors [50]. In the anterior pituitary, *DLK1* is localized in all the hormone-producing cells, supporting several regulatory functions [70]. Animal studies have revealed that *DLK1* is expressed in the normal pituitary gland, hypothalamus, spinal cord, pancreatic islet cells, and adrenal glands [50,51,52]. In *DLK1*-knockout mice, the cell-specific gene expression level of FSH was significantly reduced compared to that in wild-type mice [70]. Furthermore, *DLK1* is a paternally imprinted gene on chromosome 14q32.2 that encodes a transmembrane glycoprotein [71]. Maternal uniparental disomy and paternal deletion of the 14q32.2 region, including *DLK1*, are associated with Temple syndrome, which is characterized by short stature, hypotonia, early pubertal onset, feeding difficulties, and mild facial dysmorphism [72]. Therefore, these studies support that *DLK1*, which is a noncanonical ligand of the Notch signaling pathway, may play a role in regulating pubertal onset or progression, although the exact mechanism of its involvement is unknown.

## 9. Conclusions

The regulation of puberty is multifactorial and depends on various genetic and environmental factors. However, the pathophysiology of pubertal onset is not yet entirely understood. Although many studies have explored the genetic causes of CPP, potentially causative monogenic variants have only been observed in four genes, namely, *KISS1*, *KISS1*R, *MKRN3*, and *DLK1* [43,73].

Recent studies have reported that the Notch signaling pathway may be associated with the onset and progression of puberty. The details of the association between pubertal onset and the Notch signaling pathway remain unknown. Abnormal regulation of the Notch signaling pathway or mutations that lead to excessive or attenuated activation of the Notch pathway can result in abnormal pubertal progression. However, the available research evidence so far remains inconclusive, and more studies will be necessary to confirm or refute the suggested role of the Notch signaling pathway in pubertal development.

## Figures and Tables

**Figure 1 ijms-23-03332-f001:**
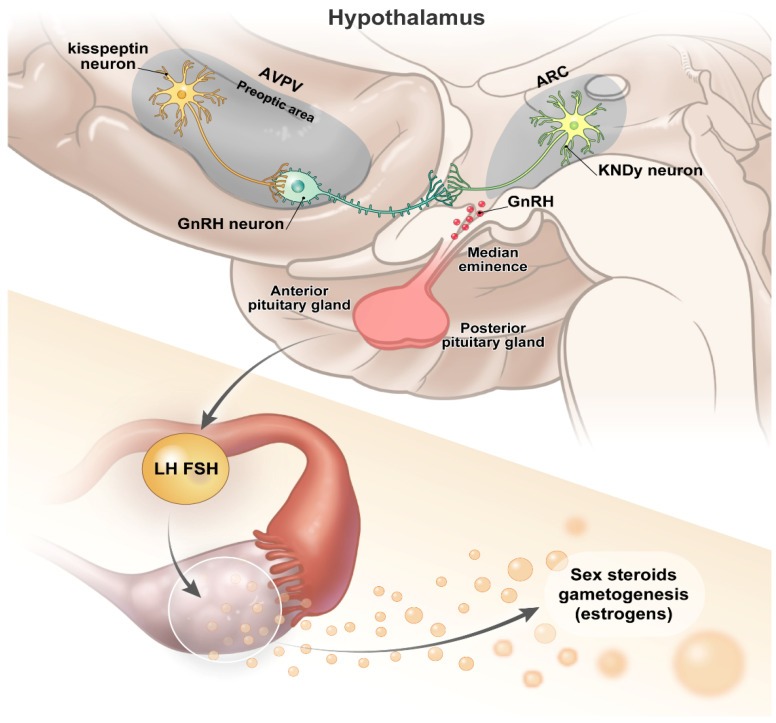
Schematic illustration of the hypothalamic–pituitary–gonadal axis. This figure illustrates the neural network regulating GnRH neurons.

**Figure 2 ijms-23-03332-f002:**
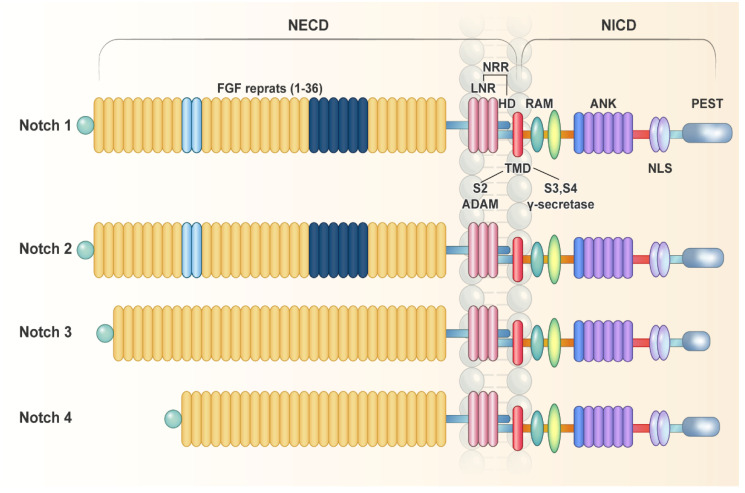
Schematic illustration of Notch receptors.

**Table 1 ijms-23-03332-t001:** Known pathogenic genes associated with central precocious puberty.

Gene	Location	Protein	Biological Function
*KISS1*	1q32	Kisspeptin	*KISS1*/*KISS1*R system plays a crucial role in the central regulation of the gonadotropic axis at puberty.
*KISS1*R	19p13.3	*KISS1*R	
*MKRN3*	15q11-q13	*MKRN3*	*MKRN3* is a ubiquitin E3 ligase that promotes the ubiquitination of target proteins.
*DLK1*	14q32	Delta-like homolog 1	*DLK1* is involved in the differentiation of several cell types including pituitary cell and adipocyte.

## Data Availability

Not applicable.

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
