# Peer review of "Aberrant Notch Signaling Pathway as a Potential Mechanism of Central Precocious Puberty"

_ijms, 2022, doi:10.3390/ijms23063332_

Round 1

Reviewer 1 Report

The authors reviewed the current knowledge about the role of the Notch signaling pathway in puberty and consider its potential involvement in the pathogenesis of central precocious puberty.

The paper is well written and it is well written. In my opinion, the manuscript is of large interest. I haven't major criticisms. I only suggest to delete the sentence "the predominant disease in girls", because it may be mis leading. 

Author Response

The authors reviewed the current knowledge about the role of the Notch signaling pathway in puberty and consider its potential involvement in the pathogenesis of central precocious puberty. The paper is well written and it is well written. In my opinion, the manuscript is of large interest. I haven't major criticisms. I only suggest to delete the sentence "the predominant disease in girls", because it may be mis leading. 

=> Thank you for your comment. We deleted that sentence as your recommendation.

Reviewer 2 Report

The manuscript by Hae Sang Lee, Young Suk Shim and Jin Soon Hwang entitled “Aberrant Notch pathway signaling as a potential mechanism of central precocious puberty” reviews the current understanding on the role of Notch signalling in precocious puberty.

My main concern is that while authors point out at the association and correlations between gene mutations and CPP, they present very little support to their case. Correlations are not causations. What is even more important is that they fail to suggest different mechanisms by which Notch signalling might affect CPP.

A particular case in DLK1 is drawn. The data is indeed intriguing, yet no working hypothesis, based on the function of this protein, is presented.

In other words, this review is not exhaustive nor beneficial to the reader, the latter ending up with more questions than answers.

In addition there are many typos and mistakes. Here some:

Authors talk about genes and proteins in the same context. Genes do not have “domains” for example. Proteins do. Genes encode proteins and while mutations are found in genes, they are transcribed to proteins, therefore the mutations are usually presented as DNA and then protein.

>…been well described that Notch signaling pathways paly fundamental….

Play

>Pp180. The exact mechanisms by which MKRN3 mutations affect pubertal onset and progression remain elucidate.

Elusive or to be elucidated

>five polymorphisms of DLK1 was identified,

were

>14q32.2 that encodes a transmembrane glycoprotein located [72].

Located where?

Author Response

The manuscript by Hae Sang Lee, Young Suk Shim and Jin Soon Hwang entitled “Aberrant Notch pathway signaling as a potential mechanism of central precocious puberty” reviews the current understanding on the role of Notch signalling in precocious puberty.

My main concern is that while authors point out at the association and correlations between gene mutations and CPP, they present very little support to their case. Correlations are not causations. What is even more important is that they fail to suggest different mechanisms by which Notch signalling might affect CPP. A particular case in DLK1 is drawn. The data is indeed intriguing, yet no working hypothesis, based on the function of this protein, is presented. In other words, this review is not exhaustive nor beneficial to the reader, the latter ending up with more questions than answers.

=> Thank you for your comment. We totally agree with you. However, there are still many unknowns about the onset and progression of puberty. Many research groups have sought to identify the genetic causes of CPP; but, to date, only four monogenic genes have been identified: KISS1, KISS1R, MKRN3, and DLK1. Given the genetic predisposition for precocious puberty, only a relatively small number of pathogenic genes have been identified. So, it is thought that more research is needed on candidate genes related to precocious puberty. Therefore, it is thought that it will be helpful for researchers to summarize the known pathogenic and candidate genes in this review journal.

In addition there are many typos and mistakes. Here some:

Authors talk about genes and proteins in the same context. Genes do not have “domains” for example. Proteins do. Genes encode proteins and while mutations are found in genes, they are transcribed to proteins, therefore the mutations are usually presented as DNA and then protein.

>…been well described that Notch signaling pathways paly fundamental….

Play

>Pp180. The exact mechanisms by which MKRN3 mutations affect pubertal onset and progression remain elucidate.

Elusive or to be elucidated

>five polymorphisms of DLK1 was identified,

were

 => Thank you for your comments. We fixed these typos. Also, we received English proofreading services by specialized company once again.

>14q32.2 that encodes a transmembrane glycoprotein located [72].

Located where?

=> Thank you for your comment. We revised this sentence.

=> Furthermore, DLK1 is a paternally imprinted gene on chromosome 14q32.2 that encodes a transmembrane glycoprotein.

Reviewer 3 Report

In this review the authors highlight the importance of Notch signalling in the regulation of puberty. The topic is interesting, but some improvement is needed.

First of all, English needs to be checked as there are few misspellings as well as some grammar mistakes with commas sometimes breaking the sentences.

Also I would suggest to revise the title by writing 'Notch signalling pathway' (and not 'Notch pathway signalling'). In addition, since an extensive part of the review also discusses about the other genes known to play a role in CPP, I would make a more comprehensive title. 

A table summarising the known mutations in the mentioned genes and the phenotypes will be good. Also it will be good to have some more biological informations on the other 3 genes, and not only on the Notch signalling. And lastly, I would introduce first the biology of the Notch signalling and then discuss about its role in puberty and the reported mutations ( so I suggest to re-organize the order of the sections).

Author Response

In this review the authors highlight the importance of Notch signalling in the regulation of puberty. The topic is interesting, but some improvement is needed.

First of all, English needs to be checked as there are few misspellings as well as some grammar mistakes with commas sometimes breaking the sentences.

 => Thank you for your comments. We fixed these typos. Also, we received English proofreading services by specialized company once again.

Also I would suggest to revise the title by writing 'Notch signalling pathway' (and not 'Notch pathway signalling'). In addition, since an extensive part of the review also discusses about the other genes known to play a role in CPP, I would make a more comprehensive title. 

=> Thank you for your comment. We changed the title as your recommendation.

=> Title is “Candidate and known genes associated with central precocious puberty”

A table summarising the known mutations in the mentioned genes and the phenotypes will be good. Also it will be good to have some more biological informations on the other 3 genes, and not only on the Notch signalling.

=> Thank you for your comment. We added Table 1.

And lastly, I would introduce first the biology of the Notch signalling and then discuss about its role in puberty and the reported mutations ( so I suggest to re-organize the order of the sections).

=> Thank you for your comment. We rearranged the order of the sections as your recommendation.